# An *in vitro* model of chronic wounding and its implication for age-related macular degeneration

Lindsay J. Bailey-Steinitz[1,2☯]*, Ying-Hsuan Shih[1,2☯], Monte J. Radeke[1], Pete J. Coffey[1,3,4,5]

1 Neuroscience Research Institute, University of California, Santa Barbara, California, United States of America, 2 Department of Molecular, Cellular, and Developmental Biology, University of California, Santa Barbara, California, United States of America, 3 Center for Stem Cell Biology and Engineering, Neuroscience Research Institute, University of California, Santa Barbara, California, United States of America, 4 The London Project to Cure Blindness, ORBIT, Institute of Ophthalmology, University College London (UCL), London, United Kingdom, 5 NIHR Biomedical Research Centre at Moorfields Eye Hospital NHS Foundation Trust, UCL Institute of Ophthalmology, London, United Kingdom

☯ These authors contributed equally to this work.
* ljbailey@ucsb.edu

**Data Availability Statement:** All sequencing data can be found in the Gene Expression Omnibus repository (GSE146884). All other relevant data are

## Abstract

Degeneration of the retinal pigment epithelium (RPE) plays a central role in age-related macular degeneration (AMD). Throughout life, RPE cells are challenged by a variety of cyto-toxic stressors, some of which are cumulative with age and may ultimately contribute to drusen and lipofuscin accumulation. Stressors such as these continually damage RPE cells resulting in a state of chronic wounding. Current cell-based platforms that model a state of chronic RPE cell wounding are limited, and the RPE cellular response is not entirely understood. Here, we used the electric cell-substrate impedance sensing (ECIS) system to induce a state of acute or chronic wounding on differentiated human fetal RPE cells to analyze changes in the wound repair response. RPE cells surrounding the lesioned area employ both cell migration and proliferation to repair wounds but fail to reestablish their original cell morphology or density after repetitive wounding. Chronically wounded RPE cells develop phenotypic AMD characteristics such as loss of cuboidal morphology, enlarged size, and multinucleation. Transcriptomic analysis suggests a systemic misregulation of RPE cell functions in bystander cells, which are not directly adjacent to the wound. Genes associated with the major RPE cell functions (*LRAT*, *MITF*, *RDH11*) significantly downregulate after wounding, in addition to differential expression of genes associated with the cell cycle (*CDK1*, *CDC6*, *CDC20*), inflammation (*IL-18*, *CCL2*), and apoptosis (*FAS*). Interestingly, repetitive wounding resulted in prolonged misregulation of genes, including *FAS*, *LRAT*, and *PEDF*. The use of ECIS to induce wounding resulted in an over-representation of AMD-associated genes among those dysregulated genes, particularly genes associated with advanced AMD. This simple system provides a new model for further investigation of RPE cell wound response in AMD pathogenesis.

within the manuscript and Supporting Information files.

**Funding:** This work was funded by the William K. Bowes Jr. Foundation, Garland Initiative for Vision (www.wkbowesjrfoundation.org), and the California Institute for Regenerative Medicine (LA1-02086, PC, www.cirm.ca.gov). The funders had no role in study design, data collection and analysis, decision to publish, or preparation of the manuscript.

**Competing interests:** The authors have declared that no competing interests exist.

## Introduction

Retinal pigment epithelium (RPE) cells are a monolayer of highly specialized pigmented cells residing between the retinal photoreceptors and Brüch's membrane. A single RPE cell maintains the health of approximately thirty photoreceptors by phagocytosing outer segments and supporting the visual cycle, among other functions [1]. As a layer of epithelium, RPE cells selectively transport substances across the blood-retinal-barrier (BRB) and secrete growth factors such as PEDF and VEGF to support the neural retina and choriocapillaris [2–4]. RPE cells play a vital role in the maintenance of retinal health; as such, degeneration of this simple layer of cells can cause an imbalance in the homeostasis of the subretinal space and may lead to permanent visual impairment [5, 6].

The loss of RPE cells is believed to be a crucial step in the onset of age-related macular degeneration (AMD), the leading cause of irreversible blindness in the elderly population of the developed world. Early stages of AMD can be identified by the presence of drusen, extracellular deposits located between the RPE and Bruch's membrane, and RPE cell abnormalities, including changes in pigmentation [7–9]. As the disease advances to later stages, it can take on two clinically distinct yet not mutually exclusive forms commonly referred to as dry and wet AMD [10–13]. Approximately 12% of early AMD cases develop into an advanced subtype of dry AMD called geographic atrophy (GA), which is characterized by the progressive degeneration of the RPE cells, photoreceptors, and choroidal capillaries near the macular region [14–18]. Additional RPE cell abnormalities associated with GA include enlarged and multi-nucleated cells at the margins of the regions of atrophy [19]. Alternatively, early AMD can progress to wet AMD, characterized by choroidal neovascularization (CNV), where neovascular tissues infiltrate the retina. Infiltration of these tissues can interfere with the RPE-photoreceptor interface leading to scarring and may leak fluid into the retina, causing further degeneration and transdifferentiation of RPE cells [20–22].

Although the exact mechanisms of AMD progression are under debate, chronic exposure to cytotoxic elements such as drusen, lipofuscin, and reactive oxygen species (ROS) can promote RPE cell death and increase the risk of AMD [23–27]. In the last decades, numerous cell-based wound healing assays, via chemical or mechanical ablation, have been developed to dissect the underlying mechanisms of RPE cell wound response and AMD pathogenesis [28–32]. Nevertheless, it is a technical challenge to create a chronic and localized wounding situation to recapitulate the progressive RPE degeneration seen in the macular region of AMD eyes. To overcome the challenge, we used the electric cell-substrate impedance sensing (ECIS) system to precisely and repetitively wound the same area on a differentiated human fetal RPE monolayer. Chronic wounding of the RPE monolayer using this system leads to significant changes in RPE morphology, behavior, and gene expression that are distinct from changes that occur after an acute wound.

## Materials and methods

### Cell culture

Human fetal RPE cells were provided by Dean Bok (University of California, Los Angeles). Fetal cells were isolated from deidentified tissue that was obtained with written informed consent by a third party tissue repository (Advanced Bioscience Resources, Alameda, CA, USA) and cultured according to previously described methods [28, 33, 34]. Cells were seeded at $1x10^5$ cells/cm$^2$ and allowed to differentiate for 32–40 days in a base medium described by Maminishkis [35]. ECIS 96-well 1E+ cultureware (Applied BioPhysics) were coated with filtered 10 mM cysteine hydrochloride (Fisher Scientific) in nanopure water for 10 minutes at

room temperature. The plate was rinsed twice with nanopure water before coating with 20 ug/ml laminin (ThermoFisher Scientific Inc.) overnight at 4˚C. Wounds were delivered using ECIS Zθ (Applied BioPhysics) with a wound current of 3000 μA, frequency of 60000Hz, and a wound time of 15 sec. Dead cells were gently removed from electrode approximately two hours post wounding by pipetting. Palbociclib (40 μM, Selleckchem); Thiazovivin (2 μM, Cayman Chemical); human recombinant TGFβ-2 (50 ng/ml, PeproTech); RepSox (50 nM, Cayman Chemical); 5-ethynyl-2'-deoxyuridine (EdU, 30 μM, Invitrogen); DKK-1 (200 ng/ml, R&D Systems); Wnt3a (200 ng/ml, R&D Systems) was supplemented to cultures on a daily basis.

### Characterization of cells

**EdU labeling and immunocytochemistry.** Proliferating cells were labeled using medium supplemented with 30 μM 5-ethynyl-2′-deoxyuridine (EdU) for 24–48 hours. Cells were fixed with 4% paraformaldehyde for 10–15 minutes. Specimens were incubated with 5% normal donkey serum at 4˚C overnight. Click-iT® Plus EdU reactions were conducted following the manufacturer's instructions (Invitrogen). Primary antibodies; Alexa Fluor 594 mouse anti-ZO-1 (7.5 μg/ml, Life Technologies), Anti-Fas clone CH11 (500 ng/ml, Sigma) were incubated overnight at 4˚C. Nuclei were stained using Hoechst 33342 (1:2000, Thermo Scientific) for 10 minutes at room temperature. ECIS wells were excised from the dish and mounded on CellVis #1.5H 12-well dishes using ProLong Gold antifade mountant (Thermo Fisher). Images were obtained using a Cytation5 (BioTek) and processed using the Gen3.0 software to produce movies. Images taken to assess cell density and morphology were taken using auto exposure.

**Transcriptomic analysis.** Cells on and in the region adjacent to the 350 μm diameter electrodes were manually dissected using a 1.5 mm biopsy punch (Integra LifeSciences). RNA was harvested using NucleoSpin RNA XS Kit (Macherey-Nagel) and converted into cDNA using SMART-Seq v4 Ultra Low Input RNA Kit (Clontech Laboratories). DNA libraries were prepared with Ion Xpress™ Plus gDNA Fragment Library Preparation kit and sequenced by an Ion Proton next-generation sequencer (Thermo Fisher Scientific Inc.). The resulting sequences were aligned to the human transcriptome and genome (hg38) using a two-stage alignment pipeline employing STAR and TMAP read aligners [36]. The number of reads per protein-coding mRNA was determined using Partek Genomics Suite (Partek Inc.), and the dataset was normalized using the trimmed mean of the M-values method [37]. Genes with reads per million (RPM) ≥1 in three or more samples were selected (S1 Table), and differential expression and statistical analysis were carried out using the classic implementation of edgeR (S2 Table) [38]. The RNA-Seq data and methods can be accessed through the Gene Expression Omnibus (GEO: GSE146884).

## Results

### Differentiated human fetal RPE cells mend lesions within 24-hours

The integrity of the RPE monolayer along with the endothelial cells of Bruch's membrane are required to maintain the blood-retinal barrier [2]. To investigate the wound-healing capacity of differentiated human RPE, electric cell-substrate impedance sensing (ECIS) Zθ technology was utilized [39–41]. In this system, cells are grown on gold electrodes located in the bottom of an ECIS cultureware plate where the electrical impedance imposed by those cells is monitored and recorded by the application of a low voltage alternating current. Discrete paddle-shaped wounds in the monolayer can be created by delivering high current-high frequency pulses for several seconds, killing the cells overlaying and directly adjacent to the electrodes (Fig 1A).

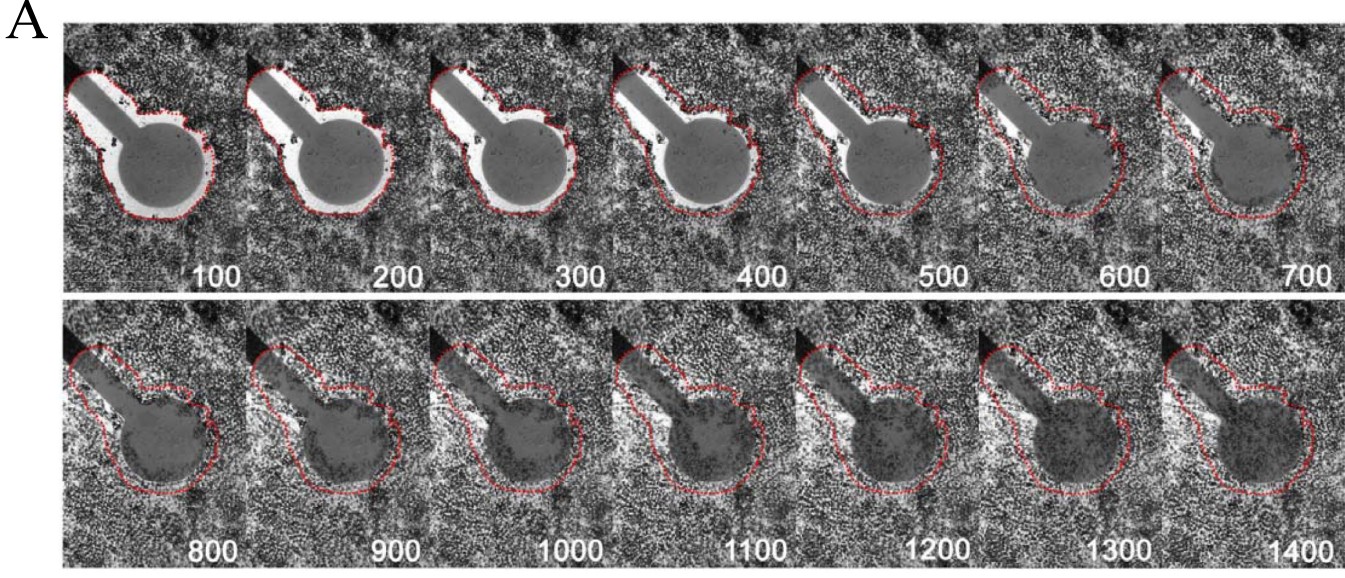

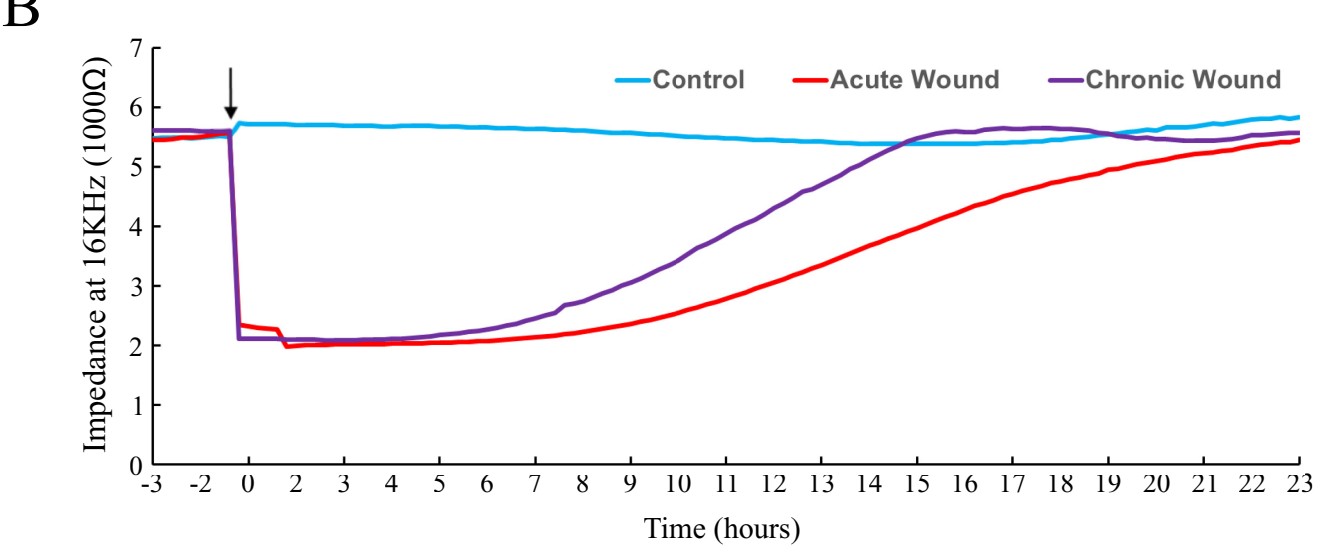

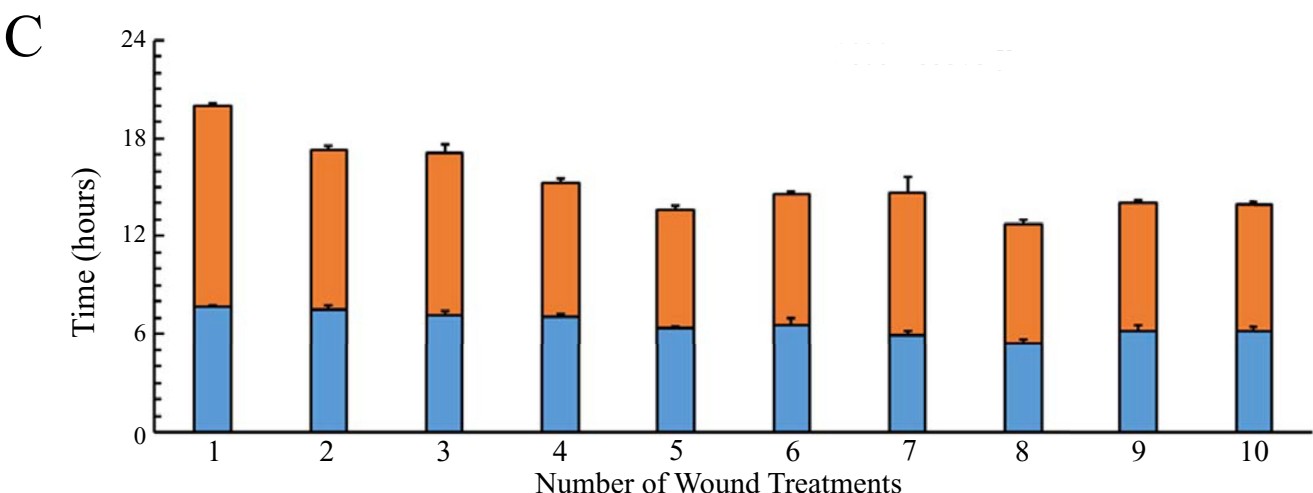

**Fig 1. RPE cell wound repair kinetics.** (A) Time-lapse phase contrast images of RPE cells wound healing. Red-dash lines indicate the original lesion border. Images were taken every 100 minutes post-wounding. Scale bar is 400 μm. (B) Real-time impedance recording of RPE wound healing. A rapid reduction in impedance occurs after delivery of a high current-high frequency electrical pulse (3 mA, 60 kHz, 15 sec.; arrow) to the monolayer. Over time, the impedance gradually recovers to a level similar to that of an intact monolayer. Impedance was plotted after the first wound for acute wound treatments (red) and after the tenth wound for chronic wound treatments (purple). Unwounded wells served as controls (blue). Trace is the average of four biological replicates. (C) Time for bystander cells to recover 10% and 90% of the maximum impedance showing a time reduction trend after daily treatments (mean ± SD, n = 4).

The kinetics of wound repair can be measured by monitoring the impedance as a function of time [42, 43].

To determine the kinetics of repair in the differentiated RPE after a single acute wound, human fetal RPE cells plated at high density and cultured for 32 days to allow for the development of cuboidal morphology and pigmentation. Differentiated RPE cells maintained an impedance at 5000–6000 Ω at a frequency of 16,000 Hz as a confluent monolayer. Immediately following the delivery of a high current electrical pulse, the impedance dropped to a level comparable to an empty electrode (~2000 Ω; Fig 1B). A lag phase was apparent after the delivery of the pulse, where the impedance of an empty electrode was maintained. Using time-lapse imaging, we determined that the lag phase consisted of two events. In the first 300 minutes, bystander cells were maintained in a latent condition, where no obvious movement was observed (Fig 1A), followed by a vigorous ingrowth of bystander cells. However, it took approximately 200 minutes for RPE cells to migrate from the perimeter of the lesion to the margin of the electrode, where changes in impedance can be detected. Notably, the majority of the RPE migrated as a sheet, while cells distal to the lesion remained stationary (S1 Movie). Following the lag phase, the impedance steadily increased to a level comparable to the unwounded monolayer within 24 hours, which was confirmed by the continuous ingrowth of RPE cells using time-lapse imaging.

**Repetitive wounding accelerates the rate of wound closure.** One advantage of the ECIS system is the capability of delivering distinct and repetitive lesions to the same geographic location in a monolayer of cells while retaining the integrity of the basement membrane (S1 Fig). This feature allows for the development of a reliable method that can model chronically wounded RPE without hindering wound healing by physical damage of the extracellular matrix. Electrical pulses were delivered to create discrete wounds in the RPE cell monolayer every 24 hours for ten consecutive days in order to evaluate the capacity of differentiated human RPE to repair during a state of chronic wounding. We monitored changes in impedance between each daily treatment as the bystander RPE cells repaired the damaged areas. The time for cells to achieve a 10% and 90% level of recovery after each treatment were used as criteria to assess the rate of RPE wound healing. After the first treatment, it took an average of 7.91 hours to regain 10% of the lost impedance and approximately 19.55 hours total to reach 90% recovery (Fig 1C). Interestingly, the amount of time to repair wound closure decreased with repetitive treatments (Fig 1B). By the tenth wound treatment, the cells regained 10% and 90% of the maximum impedance after approximately 6.69 hours (11.6% reduction) and 14.48 hours total (25.9% reduction), respectively (Fig 1C).

**Repetitive wounding promotes RPE proliferation but leads to hypotrophy of the monolayer.** To investigate whether cell proliferation is involved in the repair process of a wounded differentiated human RPE monolayer, EdU (5-ethynyl-2′-deoxyuridine) was added to label the proliferating population of cells after the last wound treatment. While an intact RPE monolayer maintained a quiescent state (Fig 2A), both EdU-positive and EdU–negative cells were observed over the round 350 μm area of the lesion, indicating RPE wound healing involves both cell proliferation and migration. The number of proliferating cells increased by nearly 2-fold in the chronic wounding condition, where cultures were wounded approximately every

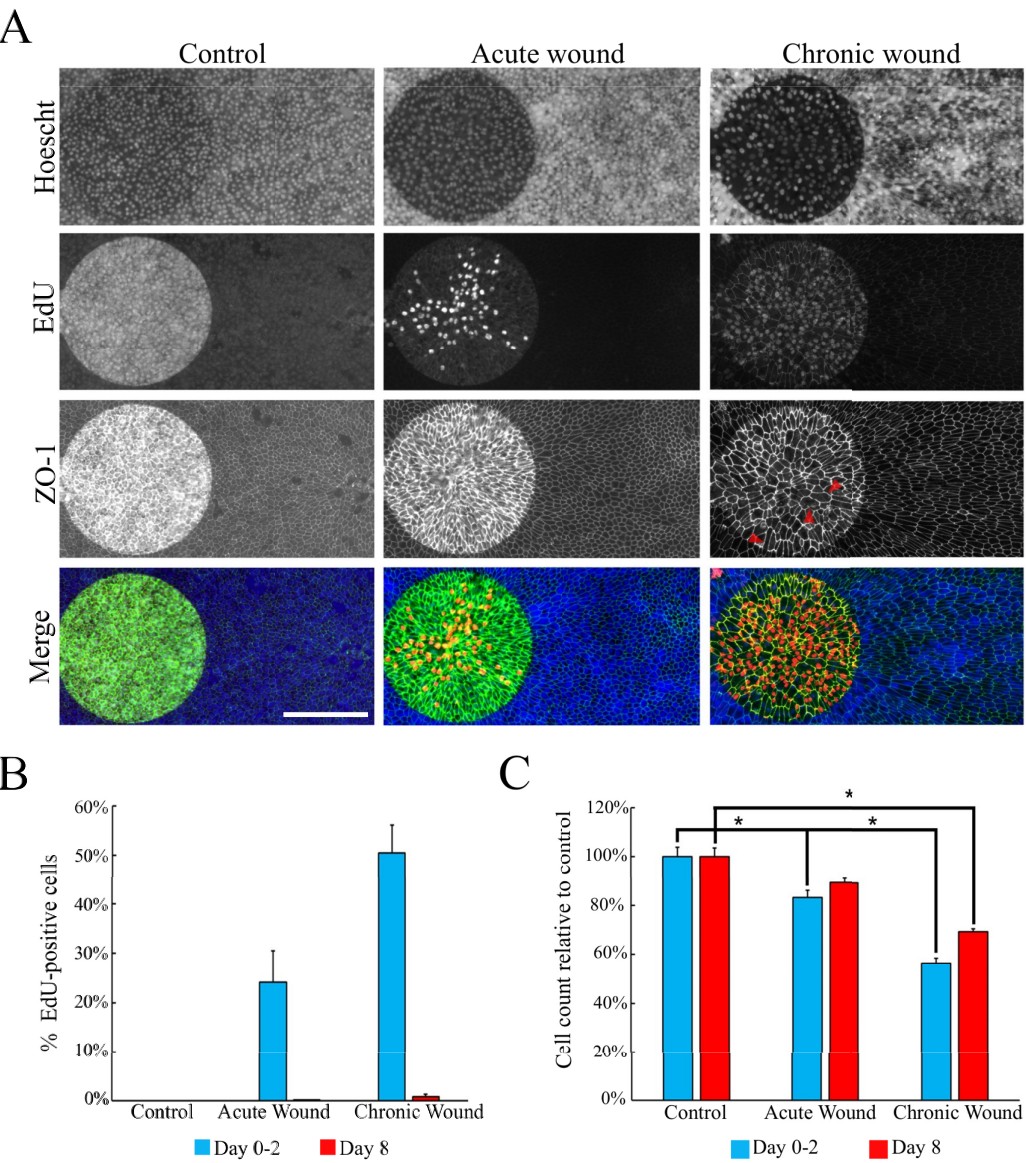

**Fig 2. RPE cells fail to regenerate fully following repeated wounding.** (A) Immunostaining of unwounded (control), acutely wounded, and chronically wounded RPE cells 48-hours after the last wound. Gold electrodes are located on the left side of each panel. Nuclei are labeled with Hoechst (blue); proliferating cells are labeled using EdU (red); cell-cell junctions are detected by ZO-1 staining (green). EdU was supplemented to the media for 48 hours post wounding. Both EdU-positive and -negative nuclei are observed in the enclosed wound, indicating that both proliferation and migration are involved in the wound closure process. Enlarged RPE cells seen after repetitive wounding are indicated by red arrows. Images were taken with auto exposure. Scale bar is 200 μm. (B) The percentage of EdU-positive cells on electrodes (mean ± SD, n = 4). (C) The numbers of cells on the electrode relative to the control (mean ± SD, n = 4, * indicates p-value<0.01, see S3 Table).

24 hours for ten days, compared to cells in the acute wounding condition, which were wounded only once (Fig 2B). Notably, the EdU-positive population was typically restricted to the enclosed wound area, whereas most of the cells outside of the lesion remained in a quiescent state. However, even after chronic wounding, cell proliferation was not sustained. The

proliferative population decreased to less than 1% of the total population in both acute and chronic wounding states by eight days after the last wound treatment (Fig 2B).

Although RPE monolayers appear to use both proliferation and migration to repair damaged areas, the RPE cells are incapable of restoring the original density after repetitive wounding. Two days after an acute wound treatment, the number of cells on the electrode was restored to roughly 85% of the control density (Fig 2C). After eight days of recovery, the cell density increased to a similar number as the intact control. However, after repetitive wounding, the regenerative ability appeared to decline as the cell number only restored to 75% of the control even after eight days of recovery. The decline in cell density resulted in enlarged RPE cells over the lesioned area, which was observed using anti-ZO-1 immunostaining (red arrows in Fig 2A).

**Inhibition of the cell cycle does not affect the rate of RPE cell wound closure.** To assess whether cell proliferation is an essential component of RPE wound closure, we blocked proliferation after wounding using palbociclib, a cyclin-dependent (CDK) 4 and CDK6 inhibitor. Supplementation of palbociclib significantly decreased cell proliferation in wounded cultures (Fig 3A and 3B) but did not affect the rate of wound closure (Fig 3D). However, there was a significant reduction in cell density over the electrode compared to chronic wound controls (Fig 3C). This data suggests that although initiation of the cell cycle is not required for RPE wound closure, the loss of proliferation results in a further reduction in cell density over lesioned areas.

**Modulation of bystander RPE cell transcriptome profile following acute or chronic wounding.** Transcriptome analysis was employed in order to gain a more comprehensive understanding of how RPE respond to acute and chronic wounding. To enrich for cells in close proximity of the wounded area, we utilized a 1.5 mm biopsy punch (red circle in S2A Fig). While electrodes in a single well encompass just 0.6% of the total surface area, a single electrode encompasses 5.4% of a 1.5 mm biopsy punch, a 9-fold enrichment. Cells were harvested at 5-hours, 24-hours, and 8-days after the final acute or chronic wound treatment, and transcriptome profiles were compiled using RNA-Seq. Acute wounding consisted of one wound treatment while chronic wounding consisted of ten consecutive wounds, once every 24-hours, to determine whether repetitive wounding resulted in any prolonged misregulation of the transcriptome. The 5-hour time point coincides with the end of the lag phase and the onset of migration. The 24-hour sample captures the point in time shortly after wound closure. The 8-day time point assesses the residual effects of wounding after the completion of proliferation and migration. Samples collected from adjacent, non-wounded cultures served as controls.

As summarized in Fig 4A, roughly 2600 genes in total were differentially expressed (FDR $\leq$ 0.05 and $\geq$ 2-fold change) compared to controls following either acute or chronic wounding treatments across all time points (S4 Table). Over 1800 differentially expressed genes (DEGs) were detected in both acute and chronic wounding conditions, while roughly 700 distinct genes remained significantly altered in acute or chronically wounded cultures alone. A majority of the DEGs were detected 5-hours after wounding. Remarkably, the expression levels of most DEGs detected in the acutely wounded 5-hour samples were restored to levels comparable to unwounded controls by 24-hours (Fig 4B). In the acute wound samples, 3.8% of the DEGs at 5-hours remained differentially expressed at 24-hours and only 0.2% genes remained differentially expressed at all time points. In chronically wounded samples there was less recovery of expression after the last wound; 17.6% of the DEGs at 5-hours remained differentially expressed at 24-hours, and 1.2% of the genes remained differentially expressed at all time points.

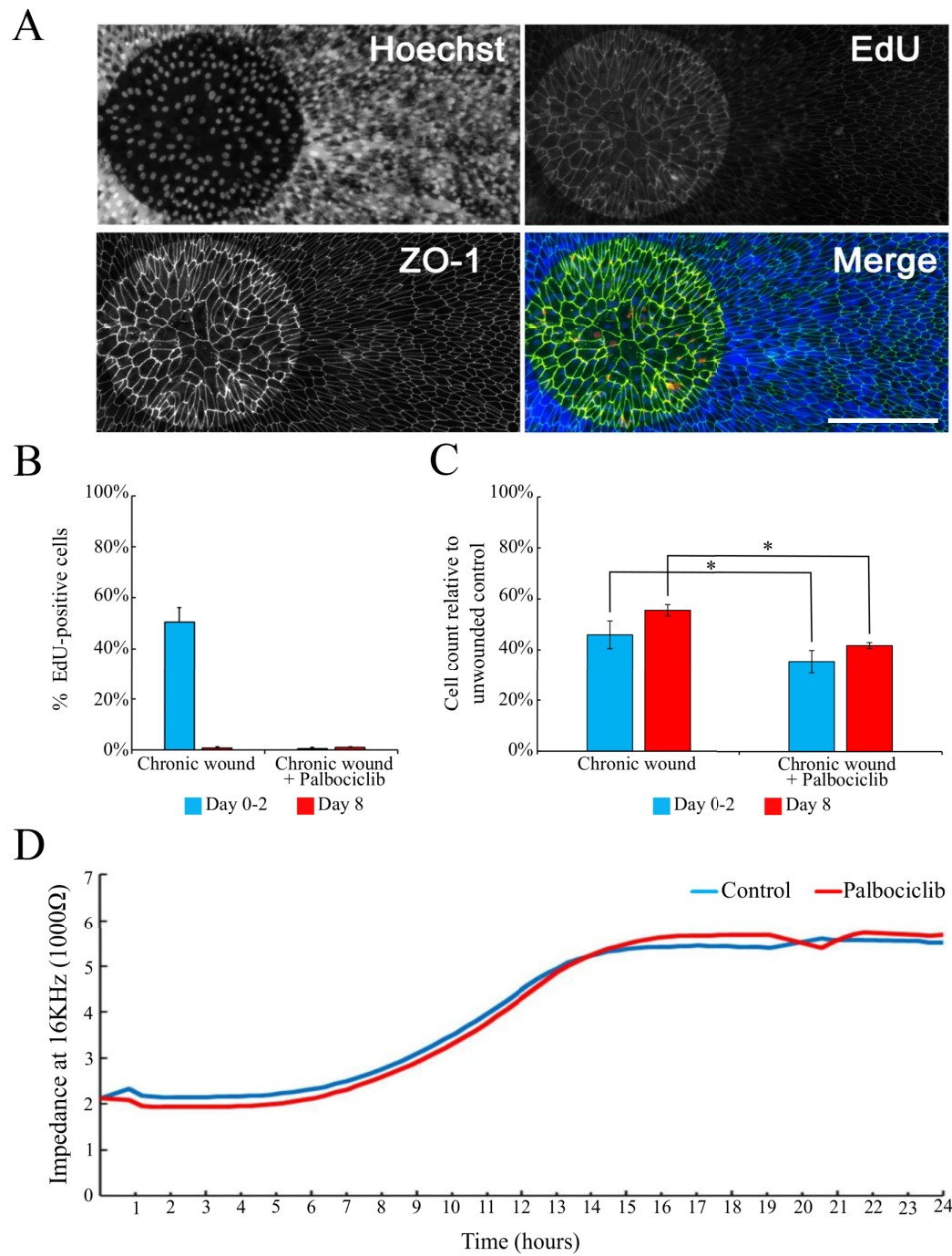

**Fig 3. The proliferation of RPE cells is not required for wound closure.** (A) Representative images of Hoescht, EdU, and ZO-1 labeled cell cultures two days after chronic wound treatments in the continual presence of 10 µM palbociclib. Scale bar is 200 µm. (B) Percent of EdU-positive cells 48-hours or 8-days after receiving chronic wounding treatments (mean ± SD, n = 4). (C) Inhibition of cell proliferation by the addition of palbociclib decreases the capability of RPE cells to restore control cell density over the electrode (mean ± SD, n = 3, * indicates p-value<0.01, see S3 Table). (D) Real-time impedance recording of RPE wound healing after chronic wound treatment in control and palbociclib treated cells. Trace is the average of three biological replicates.

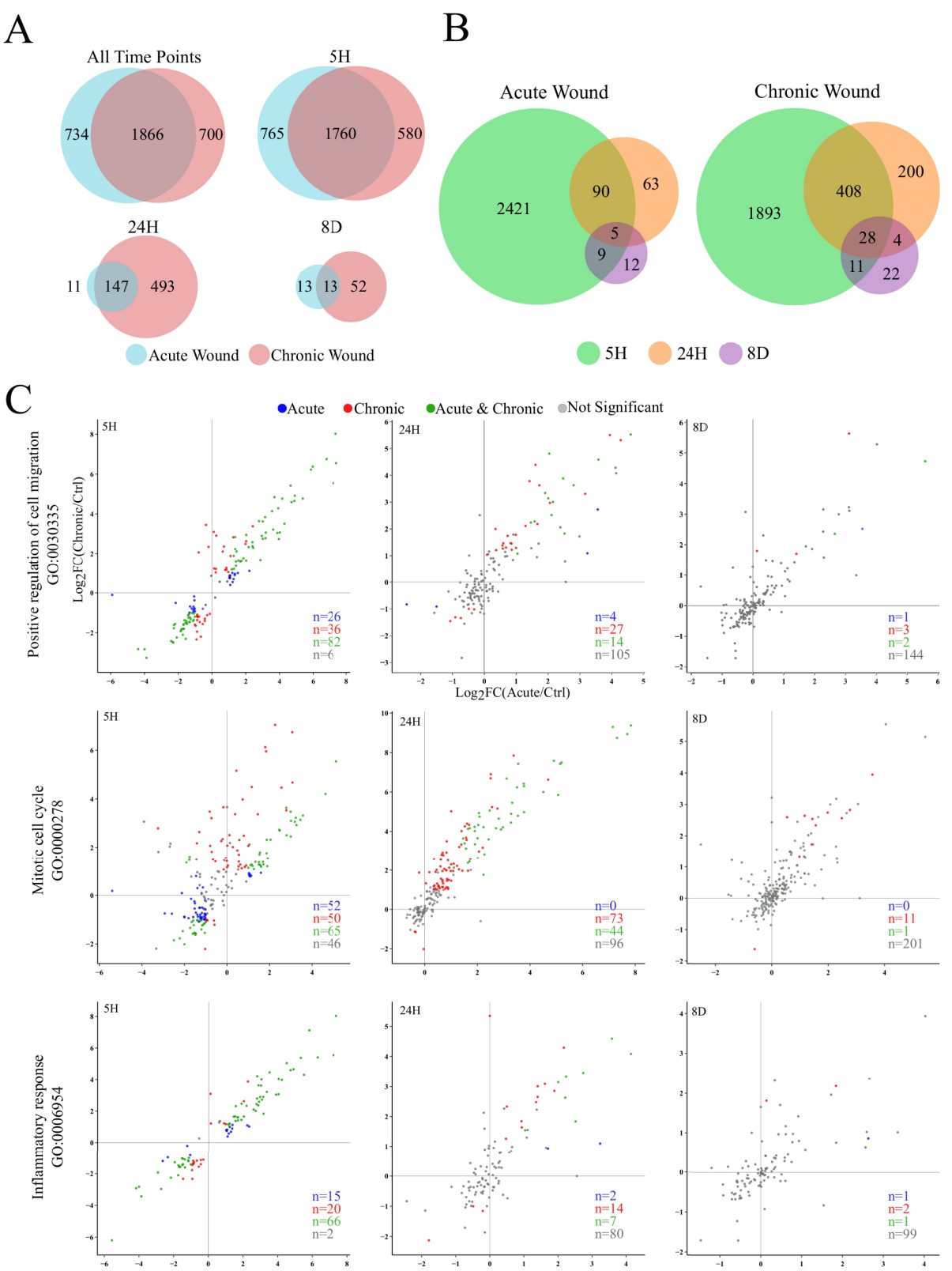

**Fig 4. Differential expression and gene ontology analysis.** (A) Venn diagrams comparing the differentially expressed genes (DEG; FDR ≤ 0.05 and ≥ 2-fold change) at three time points following acute or chronic wound treatments. (B) Venn diagrams showing the overlap of DEGs in the acute or chronic wounding condition alone after wounding. Chronic wounding results in prolonged misregulation of gene expression compared to acute wounding, as seen by the increase in the number of DEGs at both 24-hours and 8-days post wounding. (C) Scatter plots of DEGs showing $\log_2$ transformed fold change (Log$_2$FC) of acute (X-axis) or chronic (Y-axis) compared to unwounded controls at 5 hours (5H), 24 hours (24H) and 8 days (8D) after wounding. DEGs in either acute wounding alone or chronic wounding alone are colored blue and red, respectively. DEGs in both wounding conditions are colored green, and genes that are not significantly changed (either FDR > 0.05 or < 2-fold change) are colored grey. All genes are differentially expressed in one or more time point. Gene ontology groups are significantly enriched based on the total number of differentially expressed genes.

After eight days of recovery, only 26 and 65 genes remained differentially expressed in the acute and chronic wound samples, respectively. Thirteen genes remained differentially expressed in both acute and chronic wound samples after 8-days of recovery, including proteins associated with cellular structures (*ACTA2*, *TAGLN*, *KRT7*, *IQCJ-SCHIP1*), signal transduction (*DKK1*, *JUN*, *PRLR*, *RASSF3*, *WDR83*), oxidoreduction (*OGFOD2*), protease activity (*PRSS12*, *SERPINE1*), and chromatin remodeling (*SETMAR*). Of the common differentially expressed genes, the upregulation of *DKK1* stands out due to its role as a Wnt signaling antagonist, which has been shown to modulate RPE cell wound healing in a CNV model [44, 45]. However, the addition of recombinant DKK1 or Wnt3a to the culture medium did not affect the rate of wound healing or cell density of chronically wounded RPE monolayers (S3 Fig).

Using transcriptomic analysis, we showed that bystander RPE cells can rapidly adjust transcriptome profiles in response to sudden disruptions to the monolayer. Interestingly, the gene expression profile alters when the monolayer receives chronic damage compared to acute damage. For example, prolonged differential expression of genes is seen at 24-hours following chronic wounding in gene ontology groups involved in positive regulation of cell migration (GO:0030335), mitotic cell cycle (GO:0000278), and inflammatory response (GO:0006954) compared to acute wounding (Fig 4C). This observation corresponds to results showing an increased speed of wound closure and an increase in the proliferative population enclosing the lesioned area (Figs 1 and 2).

**Prolonged misregulation of key genes involved in RPE cell functions following chronic wounding.** To evaluate whether lesions on the monolayer affect the expression of key genes involved in RPE cell identity and function, we investigated the expression levels of the top 100 genes which are known to decrease in expression when RPE cells lose their epithelial identity and transdifferentiate into the mesenchymal cell fate (S5 Table) [28]. Expression levels of 81 genes were significantly altered (Benjamini & Hochberg correction, P-value<0.01 compared to controls) 5-hours post wounding, most of which have decreased expression in both acute and chronic wounding conditions (Fig 5A). All genes misregulated in the acute wound 5-hour samples were restored to control levels by 24-hours, but 26 genes remained significantly down-regulated after chronic wounding. After 8-days of recovery, expression levels of the top 100 RPE genes were not significantly different from intact control samples.

Next, we assessed the expression of genes associated with key RPE functions, including the visual cycle, growth factors, pigmentation, and retinal development. One such critical process includes the isomerization of all-*trans* retinal to 11-*cis* retinal, misregulation of which can jeopardize the visual cycle, and lead to photoreceptor degeneration and vision loss [46]. This process is carried out by LRAT, RPE65, and RDH proteins. Here, we found that the expression levels of *LRAT* and *RDH11* were significantly diminished 5-hour post wounding (Fig 5B). *RPE65* expression decreased 24-hours post wounding, but did not meet our DEG criteria. Despite the increase in expression of *LRAT* and *RDH11* 24-hours after wounding, expression levels of *LRAT* transcripts remained significantly reduced after chronic wounding, restoring to an average of less than 70% of control levels. Considering the entire lesioned area contributes

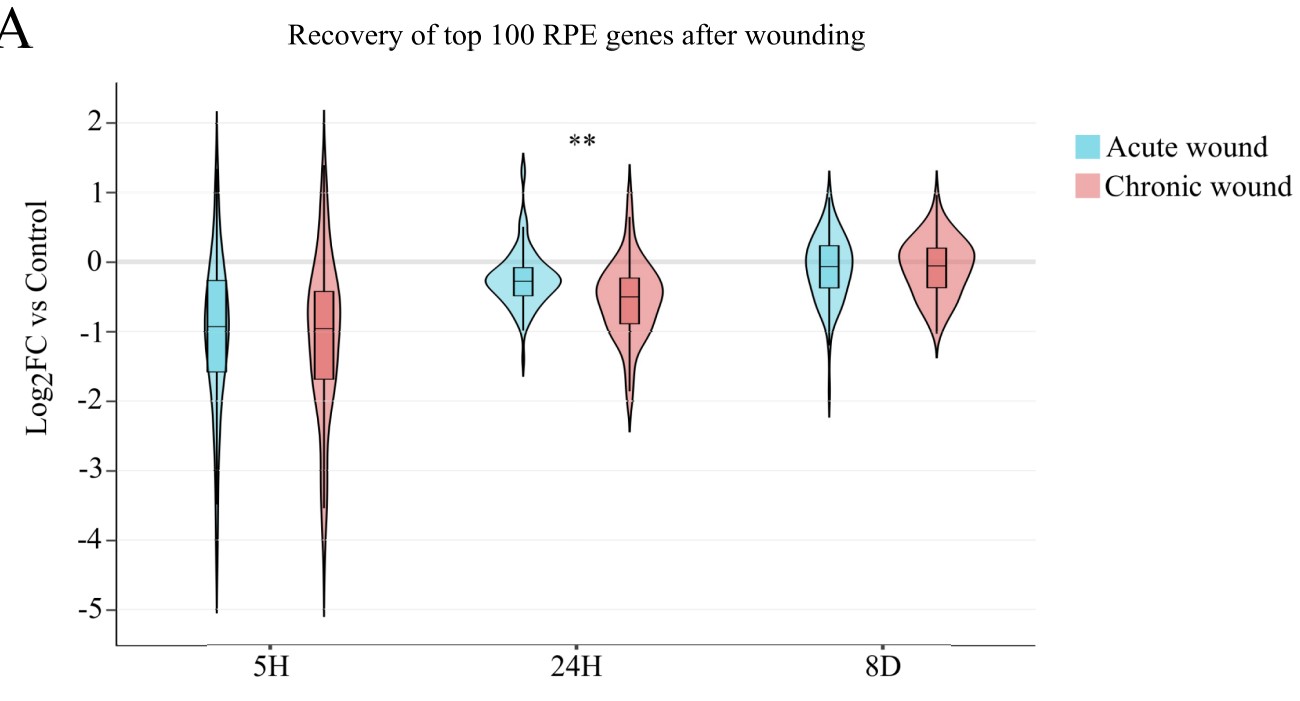

**Fig 5. Systemic misregulation of RPE genes following wounding.** (A) Violin plot showing Log₂FC of the top 100 RPE genes in acutely wounded (blue) or chronically wounded (red) RPE cells (** = p-value<0.0001). (B) Bar graphs of select RPE genes showing Log₂FC of acute or chronic wounding at 5H, 24H, and 8D compared to unwounded controls (mean ± SD, n = 3,* indicates p-value<0.01, see S3 Table).

to just 5.4% of the total area of the biopsy punch used for RNA isolation, the decrease in expression of LRAT suggests a broad decrease in expression levels across bystander cells; cells responding to the wound but not directly next to the lesion.

In addition to functioning in the visual cycle, RPE cells secrete PEDF and VEGF to support the photoreceptors and the choroid, respectively. Here, we found that the expression level of *PEDF* was minimally affected following acute wounding but significantly decreased with chronic wounding (Fig 5B). In our culture system, differentiated human fetal RPE cells express three *VEGF* isoforms, *A*, *B*, and *C*. *VEGFA* is the most abundant isoform averaging 905 RPM, *VEGFB* averages 125 RPM, and *VEGFC* is the least abundant isoform averaging 2 RPM. Previously, we have shown that *VEGFA* expression decreases while the expression of *VEGFC* increases when RPE cells terminally differentiated into a mesenchymal cell fate [28]. Here, a similar phenomenon was observed; expression levels of *VEGFA* were significantly decreased, and *VEGFC* was significantly increased in the 5-hour samples. Unlike the terminal epithelial to mesenchymal transdifferentiation seen in our previous work [28], expression levels of *VEGFA* and *VEGFC* can be restored 24 hours after wounding in acutely and chronically wounded RPE monolayers. *VEGFB* transcripts were unaffected by wounding.

Pigmentation is one of the most definitive phenotypical characteristics of RPE cells. The pigment in RPE cells can absorb scattered light to improve visual acuity and, importantly, protect retinal cells from photo-oxidative stress [5, 47]. MITF mediates the pigmentation of RPE and can also transactivate the expression of *TYR*, *TYRP1*, and *DCT*, essential enzymes for melanogenesis. Downregulation of *MITF*, *TYRP1*, and *DCT* was observed 5-hours after wounding while the expression of *TYR* was not affected. The expression levels of *MITF* and *TYRP1* restored to control levels 24 hours after wounding, but *DCT* remained downregulated in the chronic wounding condition (Fig 5B). In addition to MITF, the misregulation of other transcription factors associated with retinal development was also detected. Expression levels of *SOX9* and *CRX* decreased 5-hours after wounding and restored to control levels after 24 hours. After a single wound, the expression levels of *RAX* increased 5-hours and 24-hours after wounding but restored to control levels after 8-days. Prolonged upregulation of *RAX* was seen in the chronically wounded samples after 8-days.

Together, these results indicate that lesions on the RPE monolayer can lead to dysregulation of genes key to RPE specification and function in bystander RPE cells. Expression levels of a majority of the dysregulated genes restore by 24 hours in the acute wounding condition. However, many genes failed to fully recover after 24 hours in the chronic wounding condition, indicating that the ability of bystander RPE cells to regenerate diminishes following repetitive wounding. Due to the importance of RPE cells in maintaining the subretinal environment, prolonged dysregulation of RPE functions can potentially lead to RPE and photoreceptor degeneration.

**Association of RPE wound response with AMD pathogenesis.** In addition to genes associated with RPE cell specification and function, several DEGs are important due to their potential roles in AMD pathogenesis, particularly genes which play a role in inflammation. In this study, we observed an increase of *CCL2*, *IL-18*, and *FAS* expression in wounded samples compared to unwounded controls (Fig 6A). Expression of both *CCL2* and *IL-18* increased 5-hours after acute or chronic wounding and restored to roughly normal levels 8-days post wounding.

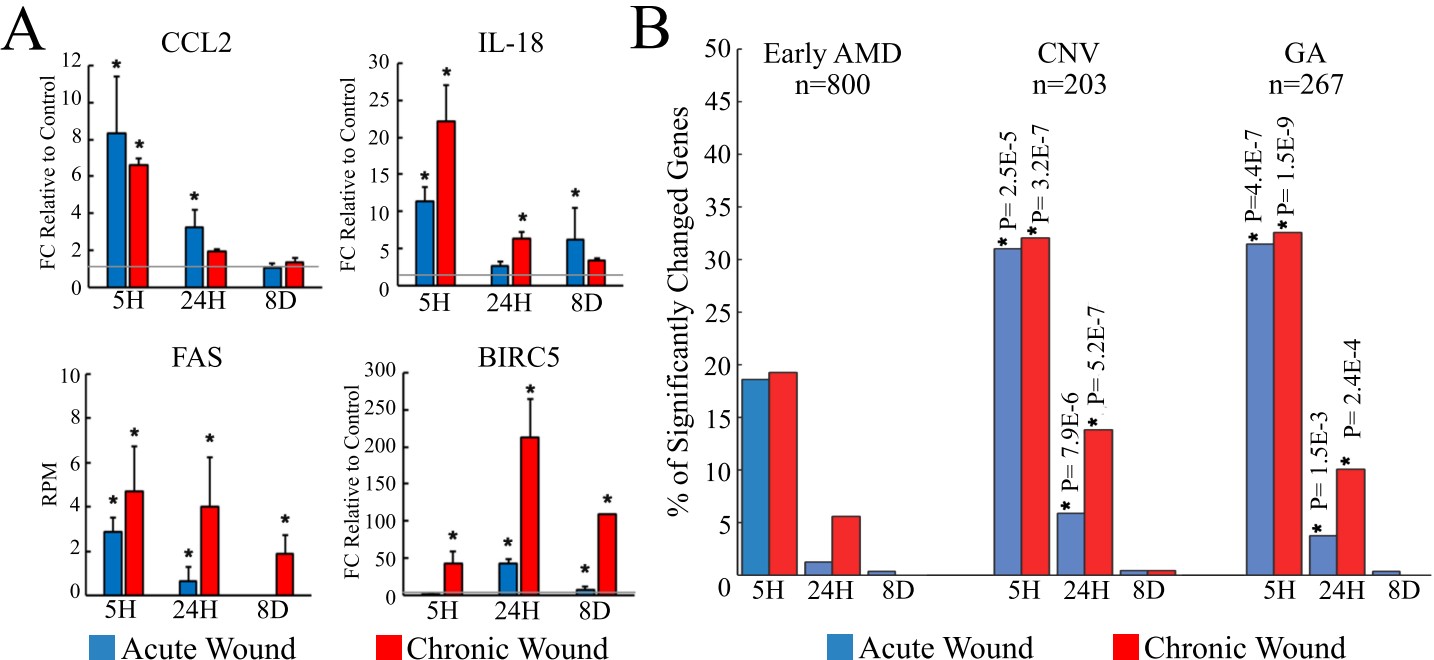

**Fig 6. Correlation between RPE cell wounding and AMD.** (A) Expression levels of selected genes. Grey line indicates the expression level of the unwounded control (mean ± SD, n = 3, * indicates p-value<0.01, see S3 Table). (B) Bar graph indicates the percentage of early AMD, CNV, and GA associated genes, whose expression levels are significantly differentially expressed (Benjamini & Hochberg correction, p-value<0.01) following acute (blue) or chronic (red) wounding. An asterisk indicates significant over-representation of CNV and GA-associated genes in wounded samples (Fisher's exact test, p-value <0.01).

Unlike *CCL2* and *IL-18*, which were expressed at detectible levels even in unwounded RPE, *FAS* transcripts were absent or in a non-detectable range in intact controls. Five-hours post wounding *FAS* was detected in both acute and chronically wounded samples (Fig 6A). In the acute wound cultures, the expression of *FAS* was reduced by 24 hours and non-detectable after 8-days. In contrast, chronically wounded RPE cells maintained upregulation of *FAS* after 8-days of recovery. Despite the confirmation of FAS expression by immunostaining, we did not observe a clear apoptotic effect on the bystander RPE cells using FAS activating IgM, as the impedance recovery profile and the cell density were comparable to unwounded controls (S3 Fig). Perhaps persistent upregulation of *BIRC5* (also known as Survivin), a member of the inhibitors of apoptosis proteins (IAPs), in the chronically wounded cells may protect bystander cells from FAS mediated cell death (Fig 6A).

Finally, we used Fisher's exact test to investigate whether using ECIS for acutely or chronically wounded RPE displays significant transcriptomic changes similar to transcriptomic profiles of AMD eyes [48]. Due to differences in the methodology, only genes expressed by *in vitro* RPE cells were considered amongst those previously detected by DNA microarray in the RPE-choroid AMD samples (S6 Table). As shown in Fig 6B, there was no significant correlation between *in vitro* RPE wounding and early AMD. Interestingly, however, a significant over-representation of genes associated with both types of advanced AMD was observed in both acute and chronically wounded samples, where chronically wounded RPE monolayers exhibited a higher correlation with both types of advanced AMD (Fig 6B).

## Discussion

In this study, we investigated the wound healing response of acutely and chronically wounded differentiated human RPE monolayers. We report that differentiated human RPE cells repair

lesions introduced by high current electrical pulses using the ECIS system and can repair repetitively induced wounds. In response to a lesion on the monolayer, bystander RPE cells migrate and proliferate to repair the wound; whereas, cells distal to the lesion remain quiescent (Fig 2A). Compared to an acutely wounded monolayer, repetitive wounding accelerates the speed of wound closure and increases the proliferative population in conjunction with prolonging the differential expression of genes related to cell migration, cell cycle, RPE function and, inflammation.

Previous reports suggest the density of RPE in the macula to be $4,960 \pm 1,040$ cells/mm$^2$, with a loss rate of 0.54% per year [49]. In our system, the density of the unwounded controls fell within previous reports, while chronic wounding resulted in a reduced cell density of ~3,000 RPE cells/mm$^2$ (S2D and S2E Fig). Despite an increase in the proliferative cell population following chronic wounding, chronically damaged RPE monolayers restore to just 75% of control density, resulting in enlarged cells over the lesion (Fig 2C). This seemingly conflicting result is likely due to the repetitive ablation of proliferative cells on the lesioned area and the lack of proliferation in the region distal to the lesion. It is possible that the RPE cells enlarge in the periphery, similar to enlarged cells seen on the electrode to compensate for cell loss while maintaining the coherence of the monolayer (S2 Fig).

Many features seen in RPE monolayers in a state of chronic wounding are strikingly similar to features seen in AMD. For instance, enlarged RPE has been reported previously in eyes with AMD, particularly near drusen [7]. The accumulation of drusen is the clinical hallmark of AMD, and it has been proposed that the presence of inflammation-associated proteins in drusen, such as complement factors, can lead to chronic immune responses in the subretinal space leading to RPE degeneration and AMD [50–53]. Using the ECIS system, we observed an increase in RPE cell size only after chronic wounding. The generation of lesions in an RPE monolayer also elicited an inflammatory response. Interestingly, the chronic wounding state showed an even more prolonged misregulation of inflammatory genes compared to the acute wounding state (Fig 4C).

In addition to the inflammatory components of drusen, mononuclear phagocytes (MPs) has been observed in both forms of advanced AMD, further supporting the idea that that chronic low-grade inflammation may play a role in the progression of AMD [54–56]. MP activation has been shown to diminish the expression of genes critical for RPE function and can induce cell death [57, 58]. CCL2 is a chemoattractant for MPs, recruiting and activating MPs to sites of CCL2 secretion. The upregulation of CCL2 expression can lead to the accumulation of MPs in the subretinal space, and CCL2 treated MPs can stimulate RPE cell apoptosis [59–62]. We observed a drastic upregulation of *CCL2* expression following both acute and chronic wounding (Fig 6A).

Apoptosis has been reported to contribute to AMD pathogenesis, particularly for RPE near drusen or GA lesions [63]. The FAS dependent apoptosis pathway is initiated by the engagement of FAS (receptor) with FASL (ligand), inducing the formation of the death-inducing signaling complex (DISC), activation of the caspase cascade, and ultimately causes DNA fragmentation [64]. Additionally, apoptosis has also been shown to be triggered in RPE cells via the IL-18 and FAS-mediated pathway, triggered by the misregulation of RNA processing [65, 66]. We found that the RPE cell response to acute or chronic wounding results in a significant increase in both *IL-18* and *FAS* expression (Fig 6A). However, expression of FASL and the components of the IL-18 receptor, IL18R1 and IL18RAP, were in the low to non-detectable range. The expression of these three components are generally expressed on immune cells such as MPs, T cells, B cells, natural killer cells, and has been reported in RPE *in vivo* [67–71].

Wounding of the RPE monolayer also caused a transient misregulation of genes key to the visual cycle, melanogenesis, growth factor expression, and RPE cell specification. Even though

bystander RPE can restore the expression levels of these genes after 8-days of repair, the capacity to recover diminishes following chronic wounding. Because of the importance of RPE cells in the maintenance of the sub-retinal environment, prolonged dysregulation of bystander RPE after chronic wounding may lead to dysfunction of the RPE monolayer leading to photoreceptor death and loss of vision [7, 19, 72].

When investigating the differentially expressed genes from the acute and chronic wounding conditions, we found a significant overrepresentation of genes that have been shown previously to be differentially expressed in late AMD eyes [48]. Similar to the RPE genes, expression levels of most of the AMD associated genes are restored after 8-days of recovery. However, a greater number of AMD associated genes were differentially expressed in the chronically wounded samples, and the extent of recovery diminished following chronic wounding.

Despite the similarity of several features of chronically wounded RPE cells and AMD, there is a fundamental difference between the model presented in this study and the advanced stages of AMD. While we observed a productive wound healing process using ECIS, even while using a cell-cycle inhibitor, a productive wound healing process is seemingly absent in advanced stages of the disease. Tissue regeneration requires the proliferation of the progenitor or bystander cells, followed by differentiation of the newly produced cells. RPE cells can reenter the cell cycle in response to growth factor stimulations such as PDGF, bFGF, TGFβ, and TNFα [73–77]. *In vitro*, primary adult and fetal human RPE cells can redifferentiate into a functional monolayer with a minimal amount of expansion. However, extensive passaging or low-density plating can direct RPE cells toward terminal mesenchymal transdifferentiation and give rise to fibrotic tissues [28]. Observations of RPE-derived fibrotic membranes in wet AMD eyes suggest that exposure to serum components may promote RPE hyperproliferation and transdifferentiation. In this study, although >50% of cells on the electrode were in a proliferative state after chronic wounding, we did not observe a clear sign of terminal mesenchymal transdifferentiation. This result is likely due to the magnitude of lesions created by the ECIS system being relatively small; therefore, an extensive propagation is not required to mend the gap.

In contrast, in GA, gross RPE proliferation and transdifferentiation are not observed. In GA, the decline in the nutrition supply due to degeneration of the choroidal capillaries together with enhanced cell apoptosis and chronic inflammation may prevent RPE regeneration. The decline in *VEGF*-A expression and the increase in the *CCL2*, *IL-18*, and *FAS* expression levels in chronically wounded bystander RPE suggest that lesions in the RPE monolayer may lead to the degeneration of choroidal capillaries and promote an inflammatory response, which can lead to further degeneration of RPE cells.

Using the ECIS system, we were able to generate an *in vitro* system to model a chronic wound state in a short amount of time with features distinct from that of an acute wounding state. However, there are limitations of this system which do not fully recapitulate the progression of AMD *in vivo*. For instance, high current is used to induce lesions by causing cell death in RPE cells overlaying the electrodes. Although this mechanism of RPE cell death is not physiologically relevant to AMD, we believe the study of the bystander wound response has potential disease-related implications, as we see several similarities between known features of AMD and chronically wounded RPE.

Another limitation of this system is the lack of underlying choroid. While the RPE monolayer maintains the heath of the overlaying RPE, the underlying choroid plays an equally important role in maintaining the health of the RPE by providing nutrients and removing waste [78]. Degeneration of the choroid, specifically the choriocapillaris, often occurs during the early stages of AMD, although the exact timing of events is still under debate [79]. Perhaps future work combining RPE, choroidal epithelial cells, photoreceptor outer segments, and one of the many types of ECIS arrays may provide more insight into how these interconnected cell

types change during wound repair. While our system does not recapitulate some aspects of an intact retina, *in vitro* models, such as this, are less expensive than *in vivo* experiments, quicker to perform, and can be efficiently scaled up.

Using ECIS as a platform for chronic wounding of RPE cells may be ideal for screening therapeutics that may enhance the ability of RPE cells to wound repair over time, which could potentially help increase the reparative capacity of RPE. In addition to the experiments presented here, this platform allows for the addition of other risk factors known to influence the onset of AMD, such as age, oxidative stress, inflammation, and mitochondrial health [80, 81]. Although we do not fully understand the pathology that drives AMD progression, this system may lead to further insights into these mechanisms. Further investigations combining chronic wounding with additional AMD risk factors may be key in further elucidating mechanisms that influence RPE wound repair in advanced stages of AMD.

## Supporting information

**S1 Movie. Real-time imaging of RPE cell wound repair.** Wounding was generated using the ECIS system. Phase contrast images were taken every 30-minutes using the Cytation5 (Bio-Tek). The video was generated using Gen3.00 software. Scale bar represents 300 μM.
(MP4)

**S1 Fig. Wounding ECIS electrodes does not affect human fetal RPE attachment.** (A) ECIS 8W10E cultureware was coated with laminin and wells were wounded once or ten times. Immediately following last wound treatment, human fetal RPE cells were plated (arrow) at 80,000/cm$^2$ to asses ability of RPE to attach to wounded electrodes. Impedance at 64,000Hz was normalized to the 1.4-hour time point, when cells were plated. Unwounded wells and a single empty well were used as controls. No significant difference in impedance between unwounded, and 1 or 10 wounds was seen, suggesting no change in the ability of RPE to attach to electrodes post wounding (n = 4).
(EPS)

**S2 Fig. Change in RPE cell size and morphology with acute or chronic wounding.** (A) Single 96-well whole mount using ZO-1 antibody to visualize cell morphology. Reflections of gold electrodes are visible. Red dotted circles indicate punch size used for RNA extraction. Solid red boxes indicate locations over the wound (w) or periphery (p). Scale bar is 1 mm (B) Morphology of unwounded RPE control cells over the electrode (w) or periphery. Scale bar is 200 μM. (C) Morphology of RPE cells over the wounded area (w) or periphery (p) at 2-days or 8-days post wounding in acute or chronic wounding conditions. Images are to the same scale as (B). (D) Cell density per mm$^2$, 2-days after acute or chronic wounding. Data was taken from Fig 2C and normalized to the area over the electrode. (E) Cell density per mm$^2$, 8-days after acute or chronic wounding. Data was taken from Fig 2C and normalized to the area over the electrode.
(EPS)

**S3 Fig. Minimal effect of Wnt3a or DKK-1 on RPE cell wound repair.** (A) Real-time impedance recording of RPE cell wound healing supplemented with DKK-1 (200 ng/ml) or Wnt3a (200 ng/ml). The recovery of impedance is not affected by supplementation with either DKK1 or Wnt3a. Each trace is an average of 2 biological replicates. (B) Cell count over the electrode based on Hoescht staining compared to unwounded samples (mean ± SD, n = 3).
(EPS)

**S4 Fig. Minimal effect of activating anti-FAS antibody on RPE cell wound repair.** (A) Immunostaining of cells expressing FAS after chronic wounding. (B) Real-time impedance recording of RPE cell wound healing supplemented with 500 ng anti-FAS activating antibody. Each trace is an average of 2 biological replicates. (C) Cell count over the electrode based on Hoescht staining relative to unwounded samples (mean ± SD, n = 2).
(EPS)

**S1 Table. Normalized RPM.** The dataset was normalized using the trimmed mean of the M-values method. Genes with reads per million ≥ 1 in three or more samples were selected for further investigation.
(XLSX)

**S2 Table. Changes in gene expression after wounding.** Differential expression and statistical analysis were carried out using edgeR. 24-hour unwounded samples were used as control for both 5-hour and 24-hour wounded samples. 8-day unwounded samples were used as the control for 8-day wounded samples.
(XLSX)

**S3 Table. P-values.** P-values for Figs 2B and 3C were calculated using a two-tailed homoscedastic student's t-test. P-values for Figs 5B and 6A were calculated using edgeR compared to unwounded controls.
(XLSX)

**S4 Table. Differentially expressed genes after wounding.** Genes with FDR ≤ 0.05 and ≥ 2-fold change compared to unwounded controls.
(XLSX)

**S5 Table. Top 100 RPE genes.** Expression levels of the top 100 RPE genes known to decrease in expression after RPE cells undergo epithelial-to-mesenchymal transition.
(XLSX)

**S6 Table. Gene list used in profiles of AMD eyes.** Genes are categorized as Early AMD, GA, or CNV and whether the expression was upregulated or downregulated in the original AMD eye profiles by Newman *et al*.
(XLSX)

## Acknowledgments

The authors would like to thank the Moorfields Eye Hospital NHS Foundation Trust for their continued support, Dean Bok and Jane Hu for materials and scientific insight, and Carolyn M. Radeke for scientific insight and review of the manuscript.

## Author Contributions

**Conceptualization:** Ying-Hsuan Shih, Monte J. Radeke.

**Data curation:** Lindsay J. Bailey-Steinitz, Ying-Hsuan Shih.

**Formal analysis:** Lindsay J. Bailey-Steinitz, Ying-Hsuan Shih.

**Funding acquisition:** Pete J. Coffey.

**Investigation:** Lindsay J. Bailey-Steinitz, Ying-Hsuan Shih.

**Resources:** Monte J. Radeke, Pete J. Coffey.

**Supervision:** Monte J. Radeke, Pete J. Coffey.

**Validation:** Lindsay J. Bailey-Steinitz, Ying-Hsuan Shih.

**Visualization:** Lindsay J. Bailey-Steinitz, Ying-Hsuan Shih.

**Writing – original draft:** Lindsay J. Bailey-Steinitz, Ying-Hsuan Shih.

**Writing – review & editing:** Lindsay J. Bailey-Steinitz, Monte J. Radeke, Pete J. Coffey.

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
