## [Decision Letter · Decision Letter 0]

28 Apr 2020

PONE-D-20-09904

An in vitro model of chronic wounding and its implication for age-related macular degeneration

PLOS ONE

Dear Bailey-Steinitz,

Thank you for submitting your manuscript to PLOS ONE. After careful consideration, we feel that it has merit but does not fully meet PLOS ONE’s publication criteria as it currently stands. Therefore, we invite you to submit a revised version of the manuscript that addresses the points raised during the review process.

Both experts who reviewed your paper were quite positive about the quality of the research and about how the paper was written.  However, both asked you to provide a summary of the gene expression results in addition to submitting the data in GEO.  Please move speculation about the significance of Figure 6 to the Discussion section.

We would appreciate receiving your revised manuscript by Jun 12 2020 11:59PM. To enhance the reproducibility of your results, we recommend that if applicable you deposit your laboratory protocols in protocols.io, where a protocol can be assigned its own identifier (DOI) such that it can be cited independently in the future. For instructions see: http://journals.plos.org/plosone/s/submission-guidelines#loc-laboratory-protocols

We look forward to receiving your revised manuscript.

Kind regards,

Alfred S Lewin, Ph.D.

Academic Editor

PLOS ONE

Journal Requirements:

Reviewers' comments:

Reviewer's Responses to Questions

**Comments to the Author**

1. Is the manuscript technically sound, and do the data support the conclusions?

Reviewer #1: Yes

Reviewer #2: Partly

2. Has the statistical analysis been performed appropriately and rigorously? 

Reviewer #1: Yes

Reviewer #2: No

3. Have the authors made all data underlying the findings in their manuscript fully available?

Reviewer #1: Yes

Reviewer #2: No

4. Is the manuscript presented in an intelligible fashion and written in standard English?

Reviewer #1: Yes

Reviewer #2: Yes

5. Review Comments to the Author

Reviewer #1: This is a well written and beautifully illustrated manuscript describing a method for injuring RPE cells either acutely or chronically, and the structural and transcriptomic changes in gene expression that accompany RPE recovery/migration/proliferation. The use of ECIS is clever, this platform is normally used for quantifying changes in cell behavior, and the authors used this system turned up to eleven to create localized RPE injury.

My only significant issue is that I recommend that, apart from the data submission to GEO, that the authors upload a spreadsheet containing the identified genes and p-values. My rationale for this is that, of cell biologists interested in the RPE, a relatively small proportion will download the raw or processed data from GEO, whereas a searchable spreadsheet with FDR values and fold changes, etc., where an investigator can look up any gene of interest to them would be more useful to a much larger number.

Minor issues or questions-

1. For figure 2, some cell shape analysis based on distance from the lesion would be interesting, a la PMID 26427486.

2. How does the RPE density in the culture system compare to the in vivo setting?

3. A scalebar would be helpful for Figure 1-3.

4. Figure 2A what do the authors make of the brighter background of EdU in the uninjured cells?

5. Do cultured RPE have gap junctions? If so does that affect the transfer of the current/size of the lesion?

6. The authors state “One advantage of the ECIS system is the capability of delivering distinct and repetitive lesions to the same geographic location in a monolayer of cells while retaining the integrity of the basement membrane.” Can they demonstrate the relative intactness of the basement membrane of the RPE in these experiments?

7. Figure 4 is an attractive and straightforward presentation of the expression data. One thing the authors may want to look at, on my pdf the dots corresponding to differentially expressed genes don’t look good when I zoom in on them, they are surrounded by a yellow box. That might go away in the final published version and just be a nuance of the pdf for review.

8. Line 267 should just be “signal transduction” I think

9. The authors evaluated 100 RPE related genes. Probably (?) the most RPE-restricted genes are RPE65 and BEST1, although their expression is often are suppressed in cell culture. Do these change expression after injury?

10. Two issues related to the culture system that are at odds with the in vivo situation is the lack of underlying choroid (which many investigators have noted is altered before RPE degeneration) and the nature of the injury (electrical destabilization) being non-physiological. The authors may want to mention these as limitations to the study.

Reviewer #2: The manuscript by Bailey-Steinitz et al developed acute and chronic RPE wound-healing models using a commercial impedance system. Authors noticed EMT-like changes in RPE cells outside the wound area. Transcriptomic analysis revealed downregulation of RPE-specific genes and upregulation of inflammatory and cell cycle genes.

Overall, the manuscript presents an interesting wound-healing model of the RPE. Wound seems quite reproducible between different experiments and cellular response is also consistent. However, the model explanation isn’t very clear. Manuscript can benefit from better data interpretation and presentation.

Is wounded area in the shape of a handle as shown in Figure 1B or a circle as shown in Figure 2A? Please clarify.

Authors do not provide any data about the status of cells after injury. Do all those cells die or only their tight junctions get damaged? Please provide staining for apoptosis and tight junction markers. Do authors wash-off dead/injured cells to allow cells from the sides to proliferate and migrate into the wound area? Please provide an explanation.

Authors argue that impedance of an RPE monolayer is established due to tight junctions between neighboring RPE cells. This statement is not supported by their results. By 1400 minutes post-wounding the monolayer doesn’t look mechanically intact, but impedance is completely recovered. This doesn’t make sense. Authors should provide IF for RPE-specific tight junction markers such as CLDN 16, CLDN 19, and E-CADHERIN to show that monolayer is fully confluent with intact tight junctions. ZO1 is not a good marker for mature tight junctions of RPE cells.

Authors argued in Figure 2 that RPE cells failed to fully regenerate the wound area after 10th round of injury. Could it be because the substrate used for RPE maturity (Laminin) was depleted? Did authors check if the culture plate maintained the same amount of coating? If authors coated the wound area after every round of injury (or every few), would cells still not recover? Please provide data that effects seen by chronic wounding are not due to Laminin depletion.

Was the monolayer impedance the same after the 10th round of injury and recovery? Please provide absolute numbers, not relative.

None of the gene expression data is provided. In supplementary data, please provide tables of all the gene expression data that was used in Figures 4-6.

In the result section for Figure 6, lot of speculation has been made. All of that speculation should be moved to the discussion part.

In general, the graph provided in Figure 6B is not supported by any data. Please provide data (in supplementary tables, if needed) and highlight which genes are common between AMD and this wound-healing model and if their expression is similarly regulated under both conditions. If such a data is not available, please remove this graph from results. Such preliminary findings may be mentioned in the discussion section.

6. PLOS authors have the option to publish the peer review history of their article (what does this mean?). If published, this will include your full peer review and any attached files.

Reviewer #1: No

Reviewer #2: No

---

## [Author Response · Author response to Decision Letter 0]

18 Jun 2020

Thank you for the time and effort put into reviewing our article entitled “An in vitro model of chronic wounding and its implication for age-related macular degeneration”. I believe your suggestions make our article stronger and data more accessible to readers. We hope all addressed comments are to your satisfaction. Listed below are the original comments with our responses, line numbers match those in the Clean version of the revised manuscript. 

From the editor-

Both experts who reviewed your paper were quite positive about the quality of the research and about how the paper was written. However, both asked you to provide a summary of the gene expression results in addition to submitting the data in GEO. Please move speculation about the significance of Figure 6 to the Discussion section.

All data has now been provided as supplemental tables and dialogue about the significance of Fig 6 has been moved into the discussion (see below for details).

Reviewer #1: This is a well written and beautifully illustrated manuscript describing a method for injuring RPE cells either acutely or chronically, and the structural and transcriptomic changes in gene expression that accompany RPE recovery/migration/proliferation. The use of ECIS is clever, this platform is normally used for quantifying changes in cell behavior, and the authors used this system turned up to eleven to create localized RPE injury.

My only significant issue is that I recommend that, apart from the data submission to GEO, that the authors upload a spreadsheet containing the identified genes and p-values. My rationale for this is that, of cell biologists interested in the RPE, a relatively small proportion will download the raw or processed data from GEO, whereas a searchable spreadsheet with FDR values and fold changes, etc., where an investigator can look up any gene of interest to them would be more useful to a much larger number.

All data has now been provided as supplemental tables as well as being accessible on the GEO database. Normalized reads are in S1 Table. All edgeR comparisons with fold changes and FDR are in S2 Table. All DEGs are provided in S4 Table. The list of the top 100 RPE genes is listed in table S5. All AMD related genes used for Fig 6 are listed in S6 Table. Lines 119, 120, 246, 293, 381.

Minor issues or questions-

1. For figure 2, some cell shape analysis based on distance from the lesion would be interesting, a la PMID 26427486.

Preliminary data suggests larger cell size after chronic wounding repair, however this experiment had different numbers of wounds compared to the article. While we saw a pretty dramatic increase in size of cells after 20 wound treatments on the electrode and up to 1000 µm away from the electrode, we did not perform the same analysis on 10 wound treatments which was used in all experiments presented here. 

2. How does the RPE density in the culture system compare to the in vivo setting?

Our control cell density falls within normal density reported by Bhatia et al., 2016 while chronically wounded cells show a much lower cell density. We have now expanded S2 Fig and added into the discussion section of the text. Lines 396-401.

3. A scalebar would be helpful for Figure 1-3.

Scalebars have been added to Figs 1-3, with corresponding changes to figure legends. Lines 152, 199, 222.

4. Figure 2A what do the authors make of the brighter background of EdU in the uninjured cells?

All nuclei (Hoescht and EdU) and morphology (ZO-1) images were taken using autoexposure. This is now stated in the materials and methods as well as in the figure legend. Lines 107, 198.

5. Do cultured RPE have gap junctions? If so does that affect the transfer of the current/size of the lesion?

Cultured RPE do have gap junctions, GJA1 is expressed in our cells, but we have not performed any through experimentation on it. It is a possibility that these gap junctions affect the overall size of the lesion. Using the ECIS system, it is also possible to change the current, frequency, and time to increase or decrease the size of the lesion. 

6. The authors state “One advantage of the ECIS system is the capability of delivering distinct and repetitive lesions to the same geographic location in a monolayer of cells while retaining the integrity of the basement membrane.” Can they demonstrate the relative intactness of the basement membrane of the RPE in these experiments?

Data has now been provided in S1 Fig with experimental detail in the figure legend. In short, we coated wells of an 8-well array, each well containing 10 electrodes, with laminin. We then wounded wells either once or ten times and immediately plated human RPE. We saw no changes in the ability of RPE to attach to the laminin coated electrodes compared to unwounded controls. Lines 163, 731-738.

7. Figure 4 is an attractive and straightforward presentation of the expression data. One thing the authors may want to look at, on my pdf the dots corresponding to differentially expressed genes don’t look good when I zoom in on them, they are surrounded by a yellow box. That might go away in the final published version and just be a nuance of the pdf for review.

The original submitted EPS file did not show the yellow boxes, but we will double check when zoomed in with the final version. 

8. Line 267 should just be “signal transduction” I think

Line 272 has been changed.

9. The authors evaluated 100 RPE related genes. Probably (?) the most RPE-restricted genes are RPE65 and BEST1, although their expression is often are suppressed in cell culture. Do these change expression after injury?

We did see changes in BEST1 and RPE65 expression, but they did not make our significance cutoff. They are both contained within the 100 RPE genes S5 Table. We now mention the expression of RPE65 in the text. Line 311.

10. Two issues related to the culture system that are at odds with the in vivo situation is the lack of underlying choroid (which many investigators have noted is altered before RPE degeneration) and the nature of the injury (electrical destabilization) being non-physiological. The authors may want to mention these as limitations to the study.

We have now addressed these limitations in the discussion section. Lines 474-491.

Reviewer #2: The manuscript by Bailey-Steinitz et al developed acute and chronic RPE wound-healing models using a commercial impedance system. Authors noticed EMT-like changes in RPE cells outside the wound area. Transcriptomic analysis revealed downregulation of RPE-specific genes and upregulation of inflammatory and cell cycle genes.

Overall, the manuscript presents an interesting wound-healing model of the RPE. Wound seems quite reproducible between different experiments and cellular response is also consistent. However, the model explanation isn’t very clear. Manuscript can benefit from better data interpretation and presentation.

Is wounded area in the shape of a handle as shown in Figure 1B or a circle as shown in Figure 2A? Please clarify.

The wound shape is the handle shape seen in Fig 1. Analysis of cell density was evaluated only over the round area of the electrode. Both have been clarified in the manuscript. Lines 130, 181-182.

Authors do not provide any data about the status of cells after injury. Do all those cells die or only their tight junctions get damaged? Please provide staining for apoptosis and tight junction markers. Do authors wash-off dead/injured cells to allow cells from the sides to proliferate and migrate into the wound area? Please provide an explanation.

Dead cells and cellular debris overlaying the electrode were removed 2 hours post wounding by pipetting. This is now stated in the Materials and Methods. Because the dead cells are removed, we believe staining for apoptosis markers is not needed. Lines 90-91.

Authors argue that impedance of an RPE monolayer is established due to tight junctions between neighboring RPE cells. This statement is not supported by their results. By 1400 minutes post-wounding the monolayer doesn’t look mechanically intact, but impedance is completely recovered. This doesn’t make sense. Authors should provide IF for RPE-specific tight junction markers such as CLDN 16, CLDN 19, and E-CADHERIN to show that monolayer is fully confluent with intact tight junctions. ZO1 is not a good marker for mature tight junctions of RPE cells.

All statements about tight junctions have been removed from the text. Lines 123-149.

Authors argued in Figure 2 that RPE cells failed to fully regenerate the wound area after 10th round of injury. Could it be because the substrate used for RPE maturity (Laminin) was depleted? Did authors check if the culture plate maintained the same amount of coating? If authors coated the wound area after every round of injury (or every few), would cells still not recover? Please provide data that effects seen by chronic wounding are not due to Laminin depletion.

We believe that laminin depletion was not the reason why RPE fail to fully regenerate after chronic wounding. After chronic wounding we see that the RPE mend the lesions faster after chronic wounding than acute wounds. In addition, we did not see any changes in the ability of RPE to attach to laminin coated electrodes after ten high current pulses. This is now added as S1 Fig. Lines 163, 731-738.

Was the monolayer impedance the same after the 10th round of injury and recovery? Please provide absolute numbers, not relative.

Impedance levels after the 10th round of injury restore to similar levels to that of the unwounded control. We have now added this to Figure 1B. Lines 152-158.

None of the gene expression data is provided. In supplementary data, please provide tables of all the gene expression data that was used in Figures 4-6.

All data has now been provided as supplemental tables as well as being accessible on the GEO database. Normalized reads are in S1 Table. All edgeR comparisons with fold changes and FDR are in S2 Table. All DEGs are provided in S4 Table. The list of the top 100 RPE genes are listed in table S5. All AMD related genes used for Fig 6 are listed in S6 Table. Lines 119, 120, 246, 293, 381.

In the result section for Figure 6, lot of speculation has been made. All of that speculation should be moved to the discussion part.

All speculation in the Fig6 section has been integrated into the discussion. Lines 352-376, 416-436.

In general, the graph provided in Figure 6B is not supported by any data. Please provide data (in supplementary tables, if needed) and highlight which genes are common between AMD and this wound-healing model and if their expression is similarly regulated under both conditions. If such a data is not available, please remove this graph from results. Such preliminary findings may be mentioned in the discussion section.

Data for Fig 6B is now provided in S6 Table. Line 381.

---

## [Editor Report · Decision Letter 1]

6 Jul 2020

An in vitro model of chronic wounding and its implication for age-related macular degeneration

PONE-D-20-09904R1

Dear Dr. Bailey-Steinitz,

We’re pleased to inform you that your manuscript has been judged scientifically suitable for publication and will be formally accepted for publication once it meets all outstanding technical requirements.

Kind regards,

Alfred S Lewin, Ph.D.

Section Editor

PLOS ONE
---

## [Editor Report · Acceptance letter]

10 Jul 2020

PONE-D-20-09904R1 

An *in vitro* model of chronic wounding and its implication for age-related macular degeneration 

Dear Dr. Bailey-Steinitz:

I'm pleased to inform you that your manuscript has been deemed suitable for publication in PLOS ONE. Congratulations! Your manuscript is now with our production department. 

Kind regards, 

on behalf of

Dr. Alfred S Lewin 

Section Editor

PLOS ONE